# Mitigating COVID-19 outbreaks in workplaces and schools by hybrid telecommuting

**Simon Mauras**[1], **Vincent Cohen-Addad**[2], **Guillaume Duboc**[3], **Max Dupré la Tour**[1], **Paolo Frasca**[4], **Claire Mathieu**[1]*, **Lulla Opatowski**[5,6], **Laurent Viennot**[7]

**1** Université de Paris, CNRS, IRIF, Paris, France, **2** Sorbonne Universite, UPMC Univ Paris 06, CNRS, LIP6, Paris, France, **3** Département Informatique, ENS de Lyon, Lyon, France, **4** Univ. Grenoble Alpes, CNRS, Inria, Grenoble INP, Gipsa-lab, Grenoble, France, **5** Université Paris-Saclay, UVSQ, Univ. Paris-Sud, Inserm, CESP, Anti-infective evasion and pharmacoepidemiology team, Montigny-Le-Bretonneux, France, **6** Institut Pasteur, Epidemiology and Modelling of Antibiotic Evasion unit (EMEA), Paris, France, **7** INRIA, IRIF, Université de Paris, Paris, France

* Claire.Mathieu@irif.fr

**Data Availability Statement:** Implementation is available at https://gitlab.inria.fr/miticov/mitigating-covid19-outbreaks All contact graphs are available at: http://www.sociopatterns.org/wp-content/

## Abstract

The COVID-19 epidemic has forced most countries to impose contact-limiting restrictions at workplaces, universities, schools, and more broadly in our societies. Yet, the effectiveness of these unprecedented interventions in containing the virus spread remain largely unquantified. Here, we develop a simulation study to analyze COVID-19 outbreaks on three real-life contact networks stemming from a workplace, a primary school and a high school in France.

Our study provides a fine-grained analysis of the impact of contact-limiting strategies at workplaces, schools and high schools, including: (1) Rotating strategies, in which workers are evenly split into two shifts that alternate on a daily or weekly basis; and (2) On-Off strategies, where the whole group alternates periods of normal work interactions with complete telecommuting. We model epidemics spread in these different setups using a stochastic discrete-time agent-based transmission model that includes the coronavirus most salient features: super-spreaders, infectious asymptomatic individuals, and pre-symptomatic infectious periods. Our study yields clear results: the ranking of the strategies, based on their ability to mitigate epidemic propagation in the network from a first index case, is the same for all network topologies (workplace, primary school and high school). Namely, from best to worst: Rotating week-by-week, Rotating day-by-day, On-Off week-by-week, and On-Off day-by-day. Moreover, our results show that below a certain threshold for the original local reproduction number $R_0^{local}$ within the network (< 1.52 for primary schools, < 1.30 for the workplace, < 1.38 for the high school, and < 1.55 for the random graph), all four strategies efficiently control outbreak by decreasing effective local reproduction number to $R_0^{local}$ < 1. These results can provide guidance for public health decisions related to telecommuting.

## Author summary

The COVID-19 epidemics has forced most countries to impose prolonged contact-limiting restrictions at workplaces, universities, schools. Using simulation and taking into

uploads/2015/09/primaryschool.csv.gz http://www.
sociopatterns.org/wp-content/uploads/2015/07/
High-School_data_2013.csv.gz http://www.
sociopatterns.org/wp-content/uploads/2018/12/tij_
InVS15.dat_.gz.

**Funding:** LO received funding from the Fondation
de France (grant 106059) as part of the alliance
framework "Tous unis contre le virus", and the
Université Paris-Saclay (AAP Covid-19 2020); C.M.
received research funding from the "Fonds
d'urgence MESRI Covid19", https://www.
enseignementsup-recherche.gouv.fr/, and from the
French National Research Agency (grant ANR-19-
CE48-0016). The funders had no role in study
design, data collection and analysis, decision to
publish, or preparation of the manuscript.

**Competing interests:** I have read the journal's
policy and the authors of this manuscript have the
following competing interests: L.O. has received
research funding from Pfizer (through her research
unit) on research related to meningococcal
epidemiology and antimicrobial resistance. All
other authors have declared that no competing
interests exist.

account the most salient epidemiological features of SARS-CoV-2, we analyze the risk of
outbreak and the impact of contact-limiting strategies on three real-life contact networks
stemming from a workplace, a primary school and a high school. The strategies investi-
gated involve (1) Rotation, in which workers are evenly split into two shifts that alternate
on a daily or weekly basis; and (2) On-Off, where the whole group alternates periods of
normal work interactions with complete telecommuting. Our study yields clear results,
whatever the studied network (workplace, primary school and high school), we find that,
from best to worst: Rotating week-by-week, Rotating day-by-day, On-Off week-by-week,
and On-Off day-by-day can all help mitigate transmission below a certain epidemicity
threshold. In the current context where institutions and companies have to quickly take
local organizational decisions and review their planning or agendas, our results should
help inform public health decisions.

## Introduction

While the world is beginning to deploy vaccinations and experimenting for more effective
cures, the COVID-19 pandemics must be contained by the deployment of suitable Non-
Pharmaceutical Interventions (NPIs), so as not to overwhelm the healthcare systems. So far,
besides mask wearing and hygiene, governments have largely resorted to generalized lock-
down orders, which have severe adverse effects on economy and society, as well as to milder
restrictions such as partial school closures, curfews, and restricting access to non-essential
businesses such as gyms and restaurants. Such NPIs and organizational adaptations have to
balance the competing goals of limiting contagion and maintaining an adequate level of
social and economic activity. Assessing the performance of containment and mitigation
strategies with respect to the propagation of the epidemic is therefore critical to making the
right policy choices and has attracted an immense research effort from many disciplines,
from medical science to economics, engineering, and social, computer and statistical sci-
ences [1–6].

Within this broad policy and research question, our work concentrates on the role of
telecommuting and how to effectively include telecommuting in the schedules of schools,
workplaces or other organizations. Our purpose is to assess and compare several telecom-
muting strategies in workplaces and schools in terms of their effectiveness in mitigating pos-
sible local outbreaks. Coming up with a precise assessment of the effects on the epidemic of
these strategies indeed requires a precise understanding of the spreading of contagion in dif-
ferent environments [7, 8]. To achieve this objective, we exploit two main ingredients: (1)
fine-grained information about contacts between individuals in different environments;
and (2) the specific behavior of SARS-CoV-2 transmission and natural history. The first
ingredient takes the form of graphs that encode daily contact networks based on (publicly
available) empirical data collected in schools and workplaces [9] (Fig 1). The second ingre-
dient is a full epidemiological transmission model for SARS-CoV-2 virus (Fig 2) that
includes the rates of contamination by individuals in different conditions, such as asymp-
tomatic or symptomatic, as well as the possible presence of "super-spreaders" [10].
Equipped with this information, one can then simulate the spread of a coronavirus epidemic
in the different work environments and evaluate the effectiveness of various mitigation
strategies. In contrast to most previous work, we focus on real-life and not on synthetically
generated contact networks.

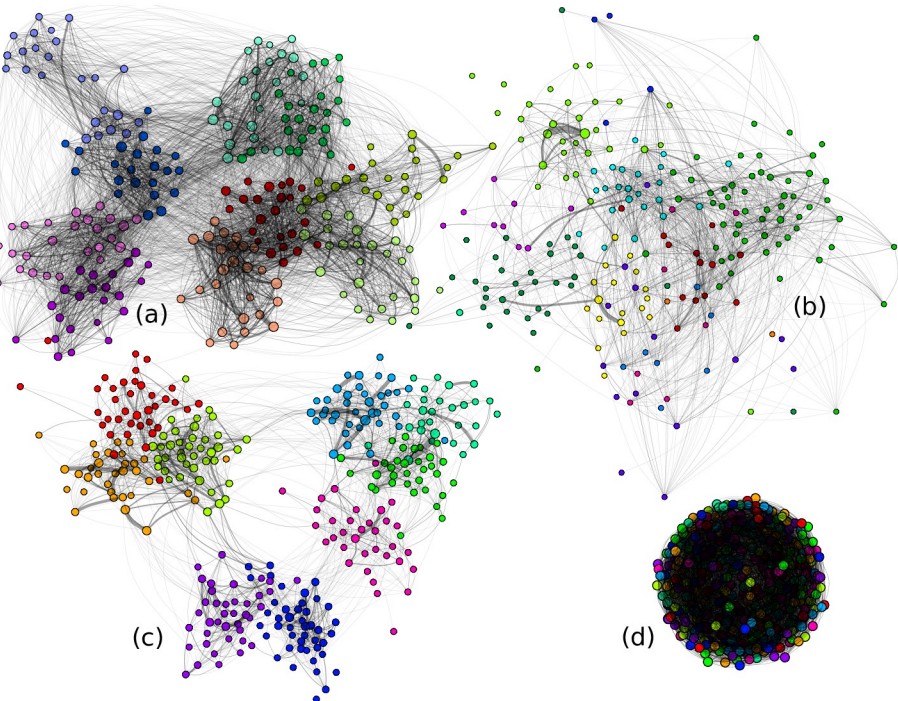

**Fig 1. Contact graphs.** (a-c) Three one-day contact networks in (a) a primary school with 242 students, (b) a workplace with 217 workers, (c) a high school with 327 students. Node colors correspond to known groups (classes or department). We see that the majority of contacts happen within groups. (d) A synthetic random graph with 9 groups selected randomly.

## Results

We simulate the spread of the virus in schools and workplaces and assess which kind of hybrid telecommuting strategy is the most effective in preventing its dissemination. For our simulations, we use fine-grained empirical data that describe person-to-person interactions and explore three contact networks collected in a primary school, a high school, and a workplace in France (Fig 1) over 2 to 10 days. We simulate the transmission of the virus over the network by implementing a stochastic transmission model of SARS-CoV-2 (Fig 2) that captures the virus clinical and transmissibility characteristics, including both symptomatic and asymptomatic individuals and super-spreading events. Several metrics are used to characterize SARS-CoV-2 transmission level following the introduction of the virus through an index case in the network in simulations: the

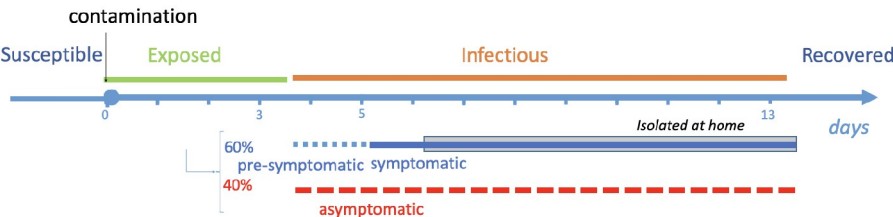

**Fig 2. The infection model for SARS-CoV-2.** The incubation (Exposed, green) lasts on average 3.7 days [11–13] and is followed by an infectious period (Infectious, orange) of 9.5 days on average, consisting of a presymptomatic period of average 1.5 days [11] followed by a symptomatic period of average 8 days [14]. We assume that symptomatic individuals self-isolate after one day of symptoms, while asymptomatic individuals (40% of the infected [15]) do not isolate.

probability that an outbreak occurs (defined here as $\geq 5$ local secondary cases), the delay until such outbreak starts, and the expected total number of locally infected individuals in case of outbreak. Assuming a baseline local reproduction number of 1.25 within the studied settings and no specific telecommuting measure implemented, the importation of the virus in the network leads to frequent local outbreaks (27% of the times for a high school, 25% for a primary school, and 26% for a workplace) and to a significant number of infections in case of outbreak, no matter the size of the studied network (34 infections on average in a high school of size 327).

In order to assess to which extent telecommuting can help mitigating the dissemination risk, five containment strategies are implemented and assessed here. Two "On-Off" strategies that consist in allowing the whole group of individuals (pupils/workers) on site (1) every other day (**On-Off Daily**), or (2) every other week (**On-Off Weekly**). We also consider two "Rotating" strategies which consist in allowing half the individuals on (1) odd days, while the other half is allowed on even days (**Rotating Daily**), or (2) odd weeks, while the other is allowed on even weeks (**Rotating Weekly**). Finally, we additionally consider the case of full-time telecommuting as a benchmark. In all these scenarios, we allow the individuals to maintain a small fraction 25% of their original local interactions even while telecommuting (thereby modeling the case of imperfect compliance by the individuals). We start the simulation with an index patient who is infected on a random day picked uniformly between 1 and 14: since the strategies are periodic with period 2 weeks, then, by symmetry, On-Off and Off-On are equivalent in our simulations.

Our results are clear: no matter which contact network they are tested on, no matter the underlying comparison metric (probability of outbreak, delay until outbreak, or expected total number of infected patients), the rankings of the four strategies are consistent (see Fig 3): the Rotating strategies significantly dominate the On-Off strategies which in turns largely dominate the absence of any policy. As expected, the full-time telecommuting (with persistent contacts only) dominates all strategies. The figure also shows that weekly and daily alternations are very similar in terms of the probability of local outbreak and of delay before outbreak, because these quantities depend on the beginning of the epidemic only; but the total number of infected people presented on the bottom panel shows that in the long run weekly alternation is a little bit better than daily alternation, both for On-Off (15.6 vs 17.4) and for Rotating (12.0 vs. 12.4) strategies. The robustness of our findings is confirmed by the extensive sensitivity analysis that we performed both on the graph structures and on the parameters of the epidemics, such as the dispersion of transmission probability, the fraction of asymptomatic patients, or variations of $R_0^{local}$. For simplicity, we only present in the main text results associated with the high school contact graph: the corresponding results for the other graphs are presented in S5–S7 Fig in the Supplementary Information.

Because the true local reproductive number in the studied settings is unknown, and to provide more insight, we study the impact of the strategies on the effective reproductive number. If $R_0^{local}$ denotes the local baseline reproductive number in the absence of strategy, what is the actual effective reproductive number $R_{effective}$ if some strategy is in place? The answer is given in Fig 4. We observe that, in the high school graph, if $R_0^{local}$ is too high (larger than 1.7), then none of these strategies, except from the full-time telecommuting, suffices to prevent the onset of an outbreak. Instead, for $R_0^{local}$ that are between 1 and 1.38, we show that all four of these strategies are satisfactory and manage to curb the epidemic. Moreover, the ranking of the strategies described above is consistent with the effectiveness of the strategies regarding the reduction of the effective reproductive number. Namely, the Rotating strategies outperform the On-Off strategies, and the full-time telecommuting outperforms the Rotating strategies.

Fig 5 provides an illustration of transmission chains resulting from the various strategies. Due to the randomness, the introduction from an infected index case within the workplace or

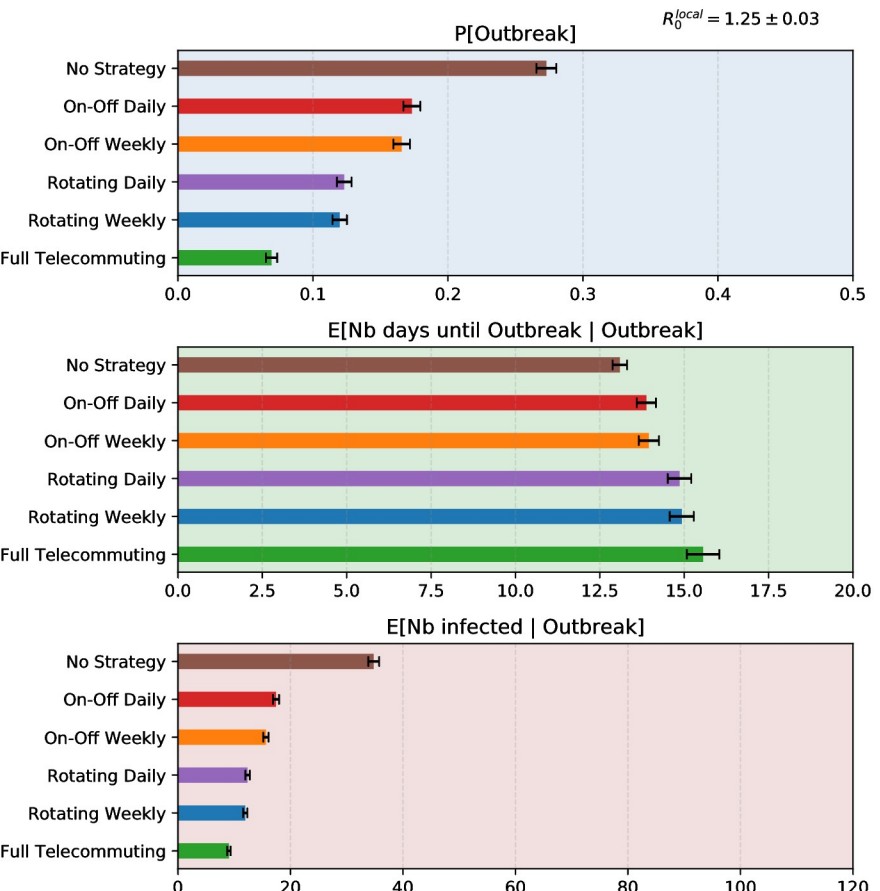

**Fig 3. Comparison of the effects on SARS-CoV-2 outbreak of containment strategies implemented in the contact graph of a high school when $R_0^{local} = 1.25$.** The three panels respectively correspond to three relevant metrics: (top) the probability that at least 5 people are infected besides patient 0 (which we define as 'Outbreak' event); (middle) the average number of days until 5 people are infected besides patient 0, conditioned on the occurrence of outbreak; (bottom) the average total number of people infected in the population in case of outbreak. That number is a random variable that has a large standard deviation, but with probability 95% its expectation lies within the error bars.

school results in a variety of propagation trees (see S19–S22 Figs). However, some general patterns are observed. First, transmissions often occur between nodes of the same color (86% of all transmissions for high school, for the baseline values of model parameters), i.e. within groups (classes in school and departments at work), reflecting the higher density of contacts within groups (93% of all contacts for high school, see Table 1 and Fig 1 and S1–S3 Figs). Second, a large share of transmissions (56%) are due to asymptomatic cases. this is due to the hypothesis that symptomatic cases isolate themselves after a day whereas asymptomatic cases do not. Third, a few super-spreading events are visible; e.g. on the top panel of Fig 5, which is sampled with no telecommuting strategy, a super-spreader event on week 4 is at the origin of 7 new branches accumulating in total 47 cases. Comparing the different trees highlights how the strategies avoid this super-spreading event and, therefore, the transmission. Indeed, for the baseline values of the parameters, averaging over all executions, in the high school contact network 15% of the tree nodes have degree at least 3 and those nodes are responsible for 61% of all infections.

We finally investigate whether our results are maintained under different contexts related to population immunization and circulating variants (see Fig 6). Even at relatively high level of

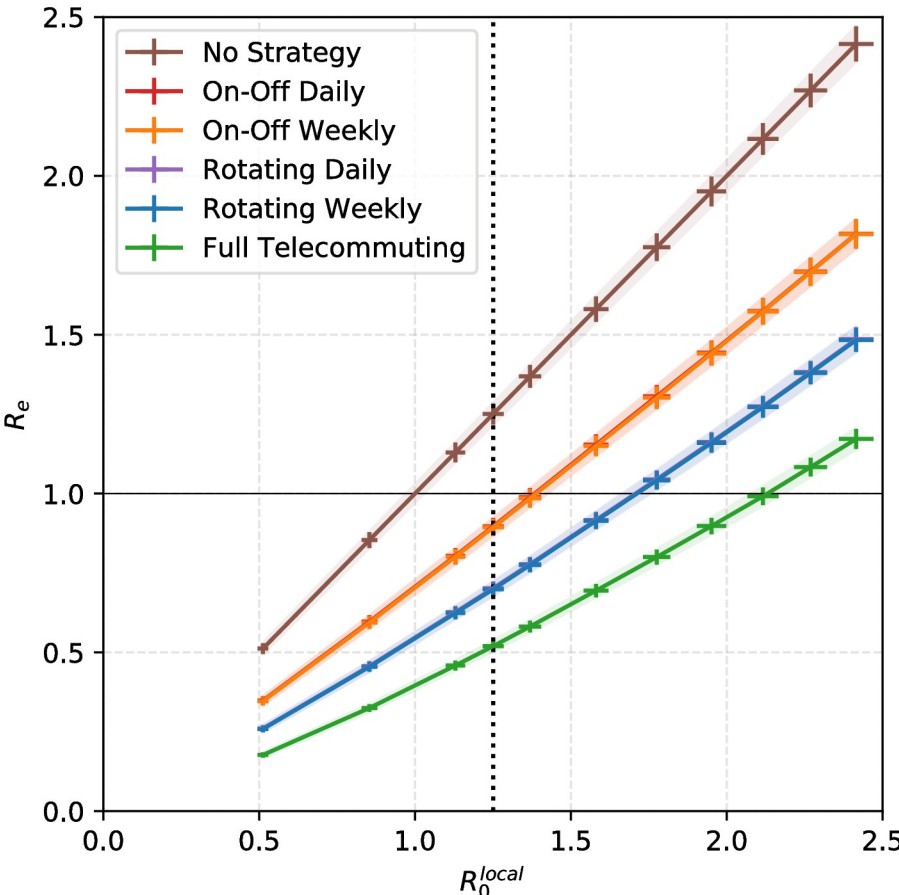

**Fig 4. Impact of the strategies for the high school contact graph.** The $x$-coordinate gives the value of the baseline reproduction number $R_0^{local}$ (mean number of persons infected by index case). For each strategy the $y$-coordinate gives the mean value of the effective reproduction number as a result of using the strategy. Thus, for our baseline value $R_0^{local} = 1.25$ (dotted vertical line), doing nothing leads to $R_e = 1.25 > 1$, whereas, as long as $R_0^{local} < 1.38$, all strategies lead to $R_e < 1$. For each curve, the shaded areas correspond to 95% confidence intervals on the estimate of $R_0^{local}$ (calculated as a function of the model parameters: horizontal error bar) and on the estimate of $R_e$ (as a function of the strategy: vertical error bar).

immunity, the introduction of a new variant with a higher transmission capacity (increase factor from 1.25 to 2) leads to a significant risk of outbreak in the "No strategy" scenario. Indeed, in such scenario, the introduction in a population that is 40% immune, of a variant 50% more infectious still leads to a probability of outbreak of 20.5%, compared to the 27.2% risk obtained in the baseline scenario with wild type virus and no immunity. Instead, we observe that the adoption of strategies significantly decreases the risk. We also investigate the impact of vaccination (or of otherwise acquired immunity), investigating a scenario where 40% of the population is partially immune and that partial immunity consists in a reduced but imperfect protection against virus acquisition and transmission. Partial immunity overall reduces the epidemic risk, but does not prevent outbreaks from happening. For example, in a situation where partially immune individuals are 50% less likely to be infected and, if infected, 50% less likely to infect others, then the probability of outbreak still equals 19.7% in the "No Strategy". Again, implementation of strategies decrease that risk. Importantly, the ranking among strategies was unchanged in all the the investigated scenarios regarding variant's transmissibility and population's immunity.

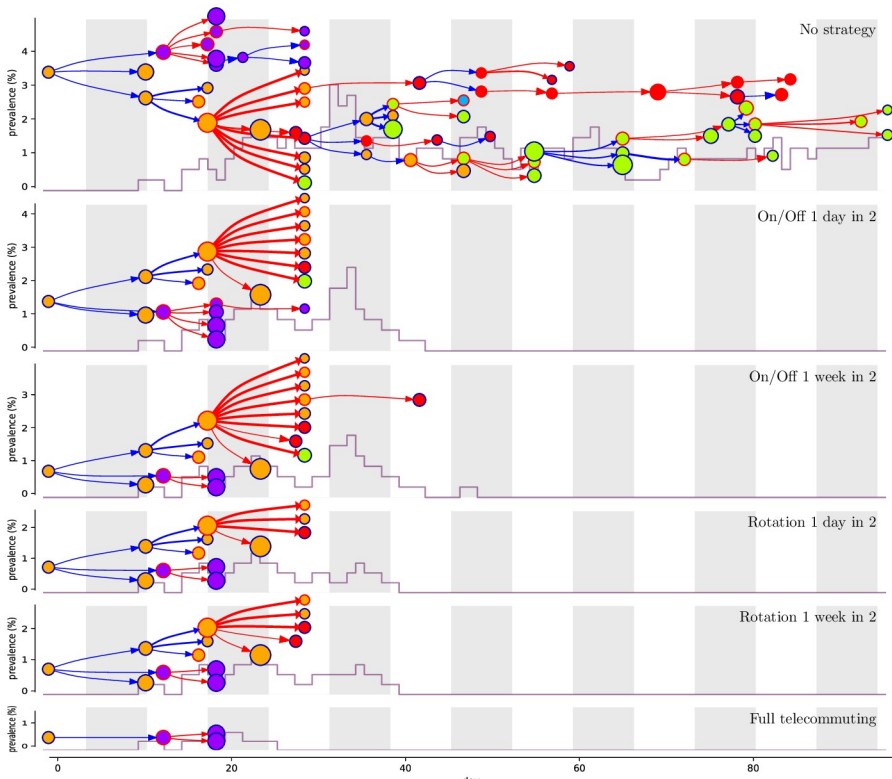

**Fig 5. Epidemic propagation in the high school contact graph for different strategies.** Each panel corresponds to an example simulation for a given strategy; strategies are sorted as by their effectiveness, from no strategy to full telecommuting. In each panel, the horizontal axis corresponds to time (day of infection) and each white or gray column corresponds to one week. The vertical axis shows the prevalence (percentage of infectious persons among the students), its evolution is plotted in grey. The epidemic propagation is shown as a tree, where each node represents an infected person and points to the persons it infects. Nodes corresponding to symptomatic (resp. asymptomatic) persons are circled in blue (resp. red). Similarly a blue (resp. red) arrow corresponds to a contamination by a symptomatic (resp. asymptomatic) person. The thickness of arrows indicates the super-spreading factor. The node color corresponds to the group of the person (class or department). The node size is linear in its degree in the graph. All the propagation trees are generated using the same realizations of the probabilistic events (run 15978 in our simulations), so that the differences between the trees are not artifacts of their randomness, but solely depend on the different strategies in place.

## Material and methods

Key elements in constructing our simulations are the choice of the contact networks and the definition of the disease transmission model, which we describe below.

**Table 1. Contact graphs characteristics.** The three studied Sociopatterns contact networks and the synthetic random graph are detailed in the table. Averaging over the days on which the data was gathered, the high school, in which data was gathered over 5 school days, comprised 327 individuals (students and teachers), each of which was in contact with 35 persons on average (degree), and the student had 230 20-second contacts per day on average (cont./pers./day). The primary school has the highest number of contacts per person in a day, followed by the high school, and finally by the workplace. All graphs have around 10 groups (classes or work departments). The percentage of intra-group contacts (perc. intra) is at least 70% in real networks while it is around 10% in a random graph with close to ten groups.

| Contact networks | nb days | nodes | degree | cont./pers./day | nb. grp. | perc. intra |
|---|---|---|---|---|---|---|
| Primary school | 2 | 242 | 68.7 | 519.7 | 10 | 72.6% |
| High school | 5 | 327 | 35.6 | 230.6 | 9 | 93.0% |
| Workplace | 10 | 217 | 39.4 | 72.1 | 12 | 76.3% |
| Synthetic graph | 1 | 327 | 35.6 | 230.6 | 9 | 10.6% |

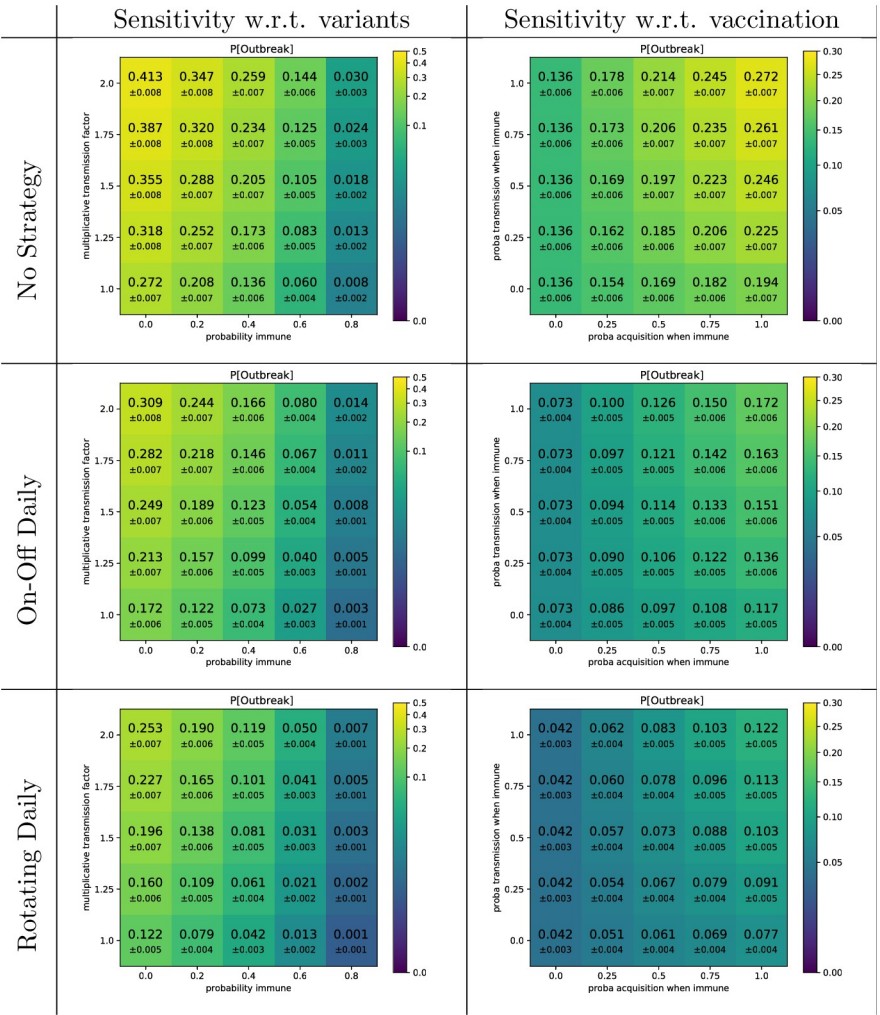

**Fig 6. Impact of strategies on outbreak probability in a partially immunized population subject to the circulation of variants, simulated in the high school contact network.** In our baseline scenario with no strategy, the probability of outbreak equals 27.2%. Each row reproduces results under specific strategies. In the left column, we study the sensitivity of this quantity with respect to potential variants showing increased transmission capacity compared to the reference strain (multiplicative factor on the *y*-axis), by assuming that such variants would circulate in partially immune populations (percentage of immune people on the *x*-axis); immune people do not get infected nor transmit the virus. In the right column, we study the sensitivity with respect to vaccination: we assume that 40% of the population is partially immune and that partially immune individuals are less likely to be infected and to contaminate others when infected. On the *x*-axis, we vary the relative probability of becoming infected and, on the *y*-axis, the relative probability of transmission by an infected partially immune individual. These results assume that patient 0 is never immune.

## Contact networks

We use traces for three different places, that are available from the SocioPatterns project (http://www.sociopatterns.org). The project collected longitudinal data on physical proximity and face-to-face contacts between individuals in several real-world environments.

1. A primary school (see [16, 17]) where 242 persons participated in 2009 over 2 days (coverage of 96% among children and 100% among teachers).

2. A workplace *Institut de Veille Sanitaire* (see [9]) where 232 employees participated in 2015 over two weeks (10 working days, coverage around 70% of the employees according to a previous deployment [18]).

3. A high school (see [19]) where 329 individuals (students) participated in 2013 over 5 days (coverage of 86% of the students in the 9 participating classes).

For each day on which data was gathered we extract a graph aggregating the data for that day: a node corresponds to an individual, an edge corresponds to a face to face contact within 1.5 meters within a 20 seconds time interval (interactions were measured using active radio-frequency identification devices (RFID)), and the weight of the edge is the number of such short contacts during the day. For comparison, we also generate a synthetic random graph, calibrated so that its main parameters (total number of nodes, of edges, and of contacts) match those of the high school contact network: more precisely, each edge is generated by selecting uniformly at random two nodes with one associated contact (rejecting loops and already generated pairs) and each of the remaining contacts is associated to an edge selected uniformly at random among the previously generated edges. Table 1 lists the main parameters of the graphs obtained by averaging over all days on which data was gathered. Fig 1 displays the three contact graphs on their first day, together with the synthetic random graph obtained. The node colors correspond to groups (classes or work departments) for the real-world contact graphs, and are chosen uniformly at random among 9 colors for the synthetic random graph. All other graphs and average graphs are depicted in S1–S4 Figs.

## SARS-CoV-2 transmission model

We model the introduction of the virus in a network by randomly sampling an index case uniformly among all the nodes to determine the patient initially infected. Similarly to related works like [20], we assume a natural history that is an agent-based model that refines classical SEIR transmission models (see Fig 2): initially individuals are *susceptible (S)*; once contaminated, having been *exposed (E)*, they go through an incubation period, after which they become *infectious (I)* after which they are assumed to *recover (R)* and develop immunity. An individual may be symptomatic or asymptomatic. In the former case, before developing symptoms she/he goes through a pre-symptomatic phase that is already infectious. We assume that transmission between an infectious and a susceptible individual happens through proximity contacts as the ones recorded in the contact network. To every 20-second contact is associated an independent small probability of transmission, $p$, so that the transmission risk increases with the duration of contact. The time step of the simulation is one day, which is consistent with the daily rhythm of commuting: as an example, if an infectious person is in contact with a susceptible person for 15 minutes during the day (and therefore through 45 "contact events"), then the probability of transmission during that day equals $1 - (1 - p)^{45}$, which for $p = 0.001$ is approximately 4.4%.

## Stochastic simulation engine

We use an agent-based model with discrete time where the time step corresponds to a day. Each person is an agent whose state is either S, E, I, or R; according to the SEIR transmission model described above. Every day, the state of each agent can change according to the number of contacts with other agents in the contact network for that day, the states of these agents, and random coin flips based on the transmission probability. The state of an agent also depends on when it had a previous state change and on what random value was obtained for the duration

**Table 2. The top table gives the parameter values used in our simulations, with the supporting references.** The bottom table summarizes some relevant quantities that can be computed from our simulations: their consistency with the literature is argued in the Discussion section.

| Notation | Description | Hypothesis | Reference |
|---|---|---|---|
| $G_d$ | graph of contacts between people that are at work on day $d$ | sociopattern graph, or random graph | Sociopatterns project [17–19] |
| $G_{ext}$ | graph of persistent contacts between people | 1/4 scaling of the average "at-work" graph | fraction of outside contacts |
| $v_0$ | patient initially infected | random uniform | null hypothesis |
| $d_0$ | day of infection of $v_0$ | random uniform | null hypothesis |
| $q$ | probability of being symptomatic | 60% | 35 to 60% [21] [22] |
| $p$ | probability of transmission during a 20-second contact | $p \leftarrow p_0 \cdot p_{sympt} \cdot p_{super}$ | |
| $p_0$ | mean transmission probability | calibrated for each graph such that $R_0^{local} = 1.25$ | |
| $p_{sympt}$ | asymptomatic relative transmission factor | 1/2 when asymptomatic, 1 otherwise | [23] |
| $p_{super}$ | super-spreading transmission factor, for each day and each person | mean = 1, Gamma(shape = 0.1) | |
| | length of exposed period | mean = 3.7, Gamma(shape = 5) | [11–13] |
| | length of presymptomatic period | mean = 1.5, uniformly 1 or 2 days | mean from [11] |
| | length of symptomatic period | mean = 8, Gamma(shape = 10) | [14] |
| | number of days of symptoms before isolation | 1 | |

| Notation | Description | Value |
|---|---|---|
| $R_0^{local}$ | average number of individuals infected by $v_0$ | 1.25 |
| $K$ | negative binomial dispersion parameter of the number of individuals infected by $v_0$ | 0.5 |
| $d_1 - d_0$ | generation interval | 5.5 when $v_0$ is symptomatic |

of the current state. A dedicated C++ program was developed for that purpose and is made available.

The model parameters are summarized in Table 2. For each infected individual, the duration of the incubation period is randomly drawn into a Gamma distribution with mean 5.2 days and shape 5 [12, 13]. The duration of the pre-symptomatic period is then uniformly drawn in {1; 2} days, consistently to published studies [11] (Table 2). The remaining duration of infectiousness follows a gamma distribution with mean 8 [14] and shape 10. We assume that the fraction of asymptomatic individuals equals 40%, within the range of [21, 22]. Symptomatic individuals are assumed to self-isolate after one day of symptoms and therefore do not cause further contamination in the studied setting; on the contrary, asymptomatic individuals stay in the system and potentially transmit the virus throughout their infectious period. The choice of the rate of instantaneous transmission is described next.

## Super-spreaders

In the COVID-19 epidemic, the number of persons contaminated by an infectious person has been suggested to show a large variance [10, 24–28]: several studies have shown that many infected individuals do not contaminate anyone, whereas a small fraction of the infected population, termed 'super-spreaders', are responsible for the majority of the transmissions. Such super-spreading events may be due to several factors including a higher viral load or infectiousness of the super-spreader, a particularly high number of contacts, and whether those contacts occur in a confined space with poor ventilation [29]. Here, we model super-spreading as follows: on each day, and for each infectious individual $i$, a random *super-spreading factor* $p_{super}$ is chosen independently from a Gamma distribution where $E[p_{super}] = 1$. Then, the transmission probability for each short contact with a susceptible individual on that day, is $p_0 p_{super}$ if $i$ is symptomatic and $p_0 p_{super}/2$ if $i$ is asymptomatic, where $p_0$ is the baseline transmission parameter.

### Calibration of the transmission parameter

The contamination parameter $p_0$ is calibrated so that the *baseline local reproduction number* $R_0^{local}$, defined as the average number of persons infected by the index case, equals 1.25. The latter value is chosen to implicitly take into account the adoption of barrier measures including social and physical distancing, mask usage and hand hygiene. The idea of inferring $p_0$ from the model is inspired by [30]. We find that $p_0 = 0.001$ in the primary school contact graph, $p_0 = 0.004$ in the high school contact graph, and $p_0 = 0.010$ in the workplace contact graph. Several values of $R_0^{local}$ were investigated, ranging from 0.5 to 2.5, corresponding (for high schools) to $p_0$ ranging from 0.001 to 0.010. The quantitative relations between $p_0$ and $R_0^{local}$ in the different graphs are reported in Fig 7.

### Persistent contacts

All simulations are initialized with an index case in the graph assumed to have been contaminated by the outside world. Importantly, we then focus on transmissions occurring within the contact graph. Since our proposed strategies act on the school or work place social networks and aims at limiting transmission clusters occurring in these specific locations, we do not model contagion of/from people who are not in the contact network. This choice is consistent with studies with similar focus [3].

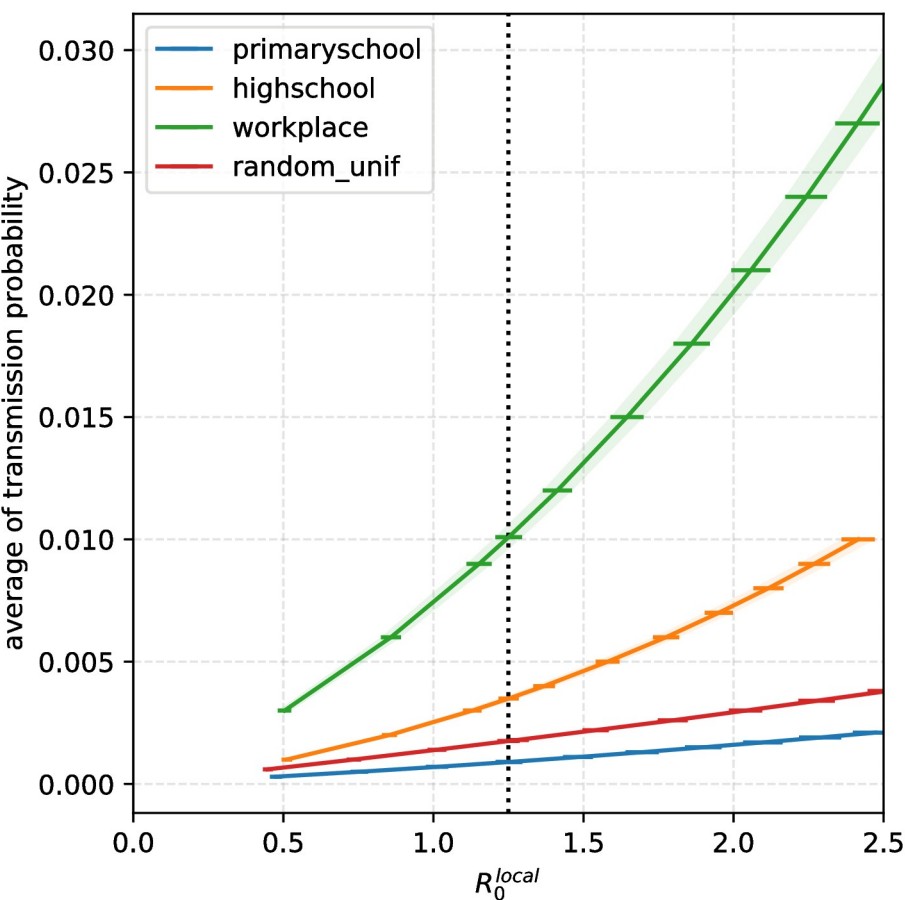

**Fig 7. Relation between the baseline transmission parameter $p_0$ and the network-dependent local reproduction number $R_0^{local}$ for our four contact networks.** We see that each curve is monotone increasing, as expected, but that the dependency is not quite linear.

Nevertheless, contacts with friends or colleagues who belong to the same social network may also happen outside the direct school/workplace environment. To model such interactions, we assume that there exists a background external graph $G_{ext}$ of *persistent contacts* that take place every day, whether workday or weekend, whether telecommuting or not. We define this external graph from the contact network by applying a dampening factor of 25% to all contacts. This factor stems from imagining a scenario in which someone would invite colleagues or fellow students to come and interact for roughly two hours during the day instead of eight hours of interaction at work (hence the 25%); and those persons would be selected from among their usual school/work colleagues, proportionally to their contacts.

## Strategies

Several non-pharmaceutical strategies were used or recommended across the world depending on activity type (school, workplace, university) or country. Here we concentrate on strategies at the level of the work/school environment which focuses on presence-sheet organization and promotion of hybrid telecommuting with partial use or partial closure of school or work environments. First, we consider **on-off** strategies, in which alternatively, either 100% of employees or students do face-to-face work, or 100% do telecommuting (distance learning). Such a strategy has, for example, been recommended as a way to exit the lockdown by alternating 4 days on and 10 days off [31]. Venezuela had for example a temporary exit strategy in which businesses were allowed to reopen on a week-on-week-off basis [32]. Second, we consider **rotating** strategies, in which 50% of employees or students do face-to-face while the other 50% do distance learning, periodically switching between the two groups. Organizing work with Rotating shifts was for example one of the actions recommended by the CDC [33]. We implement both types of strategies with different alternations: **daily** alternation (even day, odd day, not counting weekends) and **weekly** alternation (even week, odd week). Finally, we consider a **full telecommuting** strategy. This results in five strategies, which we compare in their ability to reduce the likelihood and intensity of epidemic outbreaks.

## Evaluation criteria

More precisely, strategies are evaluated based on three criteria: the probability of outbreak, defined as the percentage of simulations for which at least 5 secondary cases were infected besides the index case; the velocity of outbreak (average delay until five persons are infected); and the average cumulative number of infections until extinction of the epidemics, within outbreaks.

## Varying population immunity and cross immunity

In the current context (July 2021) vaccinations are being widely deployed in Europe and new, more transmissible, viral variants are spreading. This situation is attracting the attention of researchers: for instance, the effect of various level of immunity on the death toll in Italy has been studied in [34]. Therefore, we further evaluated the impact of the different strategies on the probability of outbreak assuming a portion of the population was already partially or fully immunized. Immunity could arise from previous natural infection or from vaccination. Since several vaccines have been introduced simultaneously and may differ in their properties, we analyse the impact of strategies under different assumptions of vaccine efficacy in terms of preventing acquisition and/or transmission. We finally assessed, in this context of a partially immune population, the remaining epidemic potential for a new strain or variant with a higher transmissibility potential.

## Discussion

### Summary

By simulating SARS-CoV-2 transmission over a diversity of contact networks, we showed how (hybrid) telecommuting reduces the virus transmission in schools and workplaces. We focused on three types of strategies: On-Off, Rotating, and Full telecommuting. Our results highlight that, whatever the contact network, these measures significantly reduce the risk of outbreak, delay the time when the outbreak occurs, and reduce the overall attack rate. This conclusion holds even though we assume some persistent contacts between individuals and a fraction of their workplace contacts, when they are not at the work location (for example, colleagues meeting outside work).

The rankings of the strategies are consistent (Fig 3): Full telecommuting (maintaining persistent contacts) significantly dominates the Rotating strategies which significantly dominate the On-Off strategies which in turns significantly dominate the absence of any policy. This strategies classification was not affected by the introductions of assumptions on vaccine- or naturally-induced immunity; nor by the modelling of an hypothetical more epidemic variant circulating in a partially immune population assuming cross-immunity.

Interestingly, despite strategies strongly differed in their probability of occurrence and size of outbreak (Fig 3), the average delay before observing an outbreak did not vary much across strategies. As a reminder, that delay-related criteria was defined as the time between index case's virus acquisition and a cumulative incidence of at least 5 secondary infected cases, conditioned on the occurrence of an outbreak. The time until that happens, conditioned on the fact that it does happen, is then largely determined by the model parameters related to virus natural history of infection, rather than by the choice of strategy. For example, considering a single index case, and an average reproductive number of 1.25, at least 2 generations of transmissions are required to reach a cumulative incidence of 5. Considering a generation time between 5 (if symptomatic) and 8 days (if asymptomatic), this would represent between 10 and 15 days. It is then expected that that delay is slightly increased when the effective reproductive number is reduced, but that it impacts more strongly the probability of outbreak.

These results are consistent with intuition. A back-of-the-envelope calculation suggests that if we order strategies according to how much they reduce contacts, the ranking is: On-off, Rotating, Full Telecommuting. Indeed, Rotating strategies always induce fewer contacts overall than On-Off strategies, because they involve the presence of smaller groups, but that does not necessarily imply less epidemic propagation because of non linear effects (in Fig 4 the relationship between $R_e$ as a function of $R_0^{local}$ are not quite linear). It is important to note that we focused here on the dissemination risk of the pathogen in a location from a single introduction by an infected individual. We therefore simulate here the beginning of an outbreak only and do not consider multiple introductions from the community.

Our results are also consistent with several studies that argue the advantage of Rotating strategies, either based on deterministic compartmental models [3, 35] or on agent-based simulations [20]. Compared with the daily alternation, the weekly alternation is naturally in phase with the duration of the incubation period and generation time of the disease. As a consequence, one can expect that it more effectively breaks the contact chains. Consistently, Fig 3 (as well as S7 and S13 Figs) show that weekly alternations are better than daily alternations.

Empirical contact data have been extensively used in infectious diseases epidemiology to realistically simulate epidemic outbreaks and assess surveillance or mitigation measures in specific environments like schools and hospitals [36–39], including the issue of school closures [16]. Regarding COVID-19 outbreak though, most previous work has focused on synthetically generated populations, both at large scale (e.g. a whole country [40, 41]) and for smaller

communities like schools [20]. Here, we build our simulations on publicly available empirical data collected in schools and workplaces [9]. Therefore, our work has some parallelism with [42], where contact data informed the agent-based simulation of COVID-19 outbreaks in a long-term care facility. Simulations on real small-scale networks enable analysing the evolution of the epidemics using realistic contact patterns, without making strong hypotheses regarding the structure of contacts and heterogeneity associated with it. As illustrated on Fig 5, transmission in such small populations is characterized by a strong stochasticity, with high chance of natural extinctions, which are less probable in epidemic models from the general population. Fig 5 also shows that due to the strong clustering, most of transmission events occur within groups rather than between group (in highschool, 93% of contacts and 86% of contaminations are within a group). Visualizing the detailed evolution of the epidemics in these environments (see Fig 5) and leveraging this fine understanding yields explicit recommendations about the effectiveness of the strategies.

A key feature of the COVID-19 pandemic is the role played by asymptomatic cases in the transmission. From the transmission trees shown on Fig 5 and S19–S22 Figs, one can note that an important proportion of the transmissions arise from asymptomatic cases. Indeed, in our baseline simulations over the high school contact network, we estimate that 56% of transmissions are due to asymptomatic individuals on average. Compared to symptomatic individuals, asymptomatic individuals are less infectious but do not self-isolate, so they have a reduced rate of transmission but over a longer period of time. The assumption that symptomatic cases would isolate relatively quickly is consistent with current recommendations in school and workplaces where individuals are asked to stay at home when they have any suspect symptoms. Imperfect compliance with isolation recommendations (which may be more realistic) would result in an increased risk of outbreak in all settings: see S14(c) Fig.

## Validity of the parameter values and robustness of the results

A lot of uncertainty exists regarding SARS-CoV-2 epidemiology and natural history and our choice of parameters, despite based on published data, deserves to be discussed. From the analysis of our simulations, we show below that our estimates are consistent with other studies.

In order to compare the risk associated with the graph structure, and not the number of contacts, we chose to analyse the three graphs by considering the same local reproduction number $R_0^{local} = 1.25$. We calibrated the baseline probability of transmission to ensure a baseline $R_0^{local} = 1.25$. Consequently, our estimates of the transmission risk over a contact strongly varied according to the analysed graph: $p = 0.001, 0.004, 0.010$ for the three graphs analysed here. Limited data is available regarding the transmission of SARS-CoV-2 in such environments and populations (here children, teenagers and adults). However, previous study on influenza transmission estimating the transmission risk from an infectious to a susceptible individual in similar contact records of 20 seconds [39] found consistent values, $p = 0.003$. The baseline value $R_0^{local} = 1.25$ was set to simulate a significant but moderate epidemic risk within the network. This value may vary due to deployment of NPIs as frequent hand washing, mask wearing and social distancing. We ran simulations for values of $R_0^{local}$ ranging 0–2.5. Higher $R_0^{local}$ values led to higher risks of outbreaks and bigger outbreak sizes (reaching 135 for $R_0^{local} = 2$), but regardless of the value of $R_0^{local}$, simulations confirm that the investigated strategies reduce the global risk compared with no strategy: all investigated strategies are able to reduce $R$ below 1 for $R_0^{local} < 1.38$ (Fig 4). Additionally variations of $R_0^{local}$ did not affect strategies classification. Overall, our results for $R_e$ (Fig 4) were comparable to [31]. Indeed, if we set $R_0^{local} \sim 1.15$, assuming that the local reproduction number is the same on weekends as on weekdays would yield $R_W = 1.48$; we then obtain that for full telecommuting $R_L = R_e \sim 0.53$,

and simulating the "4 days on, 10 days off" On-Off strategy from [31] yields $R_e \sim 0.82$. Thus, this is consistent with the findings from [31]: for $R_W = 1.5$ and $R_L = 0.6$ they find that $R_e = 0.86$. We also note that from the above calculation, our baseline intensity set at 25% for persistent contacts happens to yield a ratio $R_W/R_L$ that is almost the same as in [31], further confirming our choice of 25%.

Another key characteristics of SARS-CoV-2 dynamics is the generation time, that is, the average number of days until the secondary generation of cases are infected by the index case. Here, the effective generation time, recovered from baseline simulations of our model, is 7.3 days: this value is a weighted average of the generation times resulting from transmissions from asymptomatic individuals (8.8 days) and transmissions from symptomatic individuals (5.5 days). The latter is consistent with an estimate of 5.2 for the Singapore cluster (and a little higher than for the Tianjin cluster) [23].

Several studies have stressed the high heterogeneity in transmissions across individuals, suggesting that about 80% of transmission events are caused by only about 10% of the total cases (see [25] for example). We therefore integrated the possibility of super-spreading events in our model: even though we do not reach such high levels of dispersion, our baseline model for high schools already shows much dispersion: among all simulations, the 20% with the most secondary infections accounted for 68% of secondary infections (see S1 Text and S13 Fig).

Besides the above consistency checks, extensive sensitivity analyses were carried out to assess the robustness of our results with respect to model assumptions and parameter values: graph of persistent contacts, asymptomatic probability, 20-second mean transmission probability, asymptomatic relative infectiousness, super-spreading transmission factor, length and dispersion of exposed period, of presymptomatic period, or symptomatic period, and number of days of symptoms before isolation. They are presented in S13–S17 Figs. These sensitivity analyses show that although the evolution of the epidemic varies greatly with the parameters, corresponding variations are smooth and the ranking of the strategies is always respected. We observe that the duration of the epidemic until outbreak is the least sensitive measure, whereas the most sensitive measure is the total number of infected people when there is an outbreak (attack rate). This quantity increases when the graph of persistent contacts is replaced by a (calibrated) complete graph (S9 Fig); when $R_0^{local}$ increases (S7 Fig); and when the shape parameter of the transmission probability distribution increases, due to super-spreaders (S13 Fig). Interestingly, the attack rate also becomes much larger when the contact graph is replaced by a (calibrated) homogeneous graph (S5 Fig).

Our simulations are based on empirical data collected in three specific locations (French primary school, high school and a workplace) and are therefore specific to these locations. Nevertheless, both our results on the random network and our extended sensitivity analysis support the generality of our finding that all strategies globally reduce the dissemination risk and Rotating strategies give the best effect. Comparison with real epidemiological data in France was not possible. Indeed, in May-June 2020 variations of the Rotating and On-Off strategies were implemented in most schools, but at that time tests were not readily available to measure the impact of those measures and the virus was nearly absent. However, assessing the impact of these strategies on transmission risk in schools should be possible in the near future, because tests are now widely available and used while some regions have high levels of viral circulation.

## Limitations of our study

The results presented here should be interpreted in the light of our rather simple assumptions.

Firstly, virus transmission is assumed to occur within the contact network only, thereby neglecting potential acquisitions through external contacts, such as family members or friends are not considered, with the exception of the index case. As a matter of fact, when the levels of community circulation of the virus are high, individuals can also be exposed to the virus outside school or work place, this chance being potentially increased over telecommuting periods. Our objective was not to provide predictions about the expected prevalence in schools or workplaces but rather to evaluate the virus dissemination risk *within the network*, or in other words, the network vulnerability: we therefore focused on the quantification of this risk following a single introduction of the virus by an index case. Other studies [20] that include repeated acquisition from the community and simulate the epidemics evolution over longer time spans have reached conclusions that are consistent with ours, namely, on the advantage of rotating strategies.

Conversely, infected people within the social network might in turn infect members of other social groups, but those are outside the reach of the proposed strategies: this effect was not analyzed here. Nevertheless, in order to more realistically model contact patterns in a situation where telecommuting recommendations are not strictly enforced or complied with, we assumed that telecommuting individuals maintain a fraction of their in-network contacts. This could depict a situation where compliance with telecommuting recommendations is low. In a case of a strict lockdown or curfew, individuals would have no contacts over their telecommuting periods and may not be able to visit colleagues or school friends. In terms of modelling, this scenario simply requires removing persistent contacts and leads to a lower risk of outbreak than the one presented in our baseline model. In order to assess how changes in persistent contacts network affect the reported results, we simulated the model for different assumptions related to the external persistent graphs. The top left panel of S9 Fig shows that when a strict lockdown removes all persistent contacts (that situation is obtained in the simulation by multiplying the external contact graph by a factor of 0%), the outbreak probability drops from 27% to 17%. Moreover, in that case, all of our strategies cause the reproduction factor to drop below 1 for all values of $R_0^{local} < 1.6$ and for all contact networks (see S11 and S12 Figs). Thus, adding a curfew on top of a hybrid telecommuting strategy leads to significant improvements.

Secondly, we performed our simulations on a small set of empirical contact graphs that were built from publicly available data about just three schools and workplaces. Extrapolating from such a small set should be done cautiously. However, we are comforted by two facts. First, the main features of those networks, such as their degree distribution and their community structure (more than 70% of contacts within classes or departments; see Table 1), are representative of the social groups that we are interested in. These features are indeed key to shaping the progression of the epidemics: our simulations therefore allow studying mitigation in networks with such community structure. Second, despite significant differences between the three empirical graphs, our results are consistent across all of them and are robust to variations in the simulation parameters. Furthermore, our qualitative conclusions also extend to synthetic random graphs as described in Materials and Methods. However, quantitative results for the random graph are rather different from the original graph that has been used to tune its parameters (see S5 Fig): for instance, estimated risks were significantly worse in terms of total attack rate but better in terms of outbreak explosion. This discrepancy cautions against deriving quantitative predictions from homogeneous random models and highlights the importance of using empirical data and heterogeneous graphs that take into account real contact structures when assessing virus transmission. Another important aspect is that population coverage was imperfect in the Sociopatterns experiments. Indeed, information is missing for 4% (children), 14% (students), and 30% (employees) of the nodes of the contact networks. In

order to assess to which extent data incompleteness could affect our results, we run a sensitivity analysis in which we artificially decreased the number of participants in each graph to remove a random fraction (their connections are lost and they do not participate anymore to the transmission). Corresponding results are presented in S18 Fig. Whatever the incompleteness level, strategies order is preserved. Lower participation rates lead to underestimating the epidemic risk and the epidemic size while overestimating the delay. This analysis confirms that, up to a certain threshold, the fact that Sociopatterns contacts graphs were not perfectly observed should not affect our main conclusions.

Thirdly, for simplification, we assumed in our baseline model a fixed probability of being asymptomatic, equals to 40% whatever the population used. Studies have suggested that children are likely to be more frequently asymptomatic than older individuals. To address that point, we ran a sensitivity analysis varying the probability of being asymptomatic in the different networks. Results suggest that our main results were not affected by this hypothesis, see S15 Fig.

## Implementation and choice in practice

Of course, the choice of strategies also crucially depends on criteria such as the feasibility in practice, ease of implementation, etc. For example, hybrid teaching, in which teachers have to manage distance-teaching for half their group and onsite-teaching for the other half, has been used in many French universities since the beginning of the epidemic, but it may be more convenient for an instructor to teach on and off, having the full group either online or onsite. On the other hand, in sectors such as manufacturing a minimum of workers on site may be essential to maintain production, and then Rotating will be the most appealing strategy. We note that the main ingredient differentiating Rotating from On-Off is the breaking of connections between groups (except for the persistent contacts described above). A strategy in a school that would, for example, bring in all students of one level on even weeks and all students of another level on odd weeks would resemble On-Off more than Rotating, because it would not break the groups of students who are in contact.

It is also important to note that modifications in the implementation of the proposed strategies could result in potentially different dynamics. Let's discuss possible consequences of two natural variations of the proposed strategies. In the first one, a daily Rotating strategy is implemented but schedule is set such that every employee meets each colleague at least once a week. This strategy generates leaky isolation of subgroups, and is therefore expected to limit control efficiency. In the second one, Rotating is planned so that collaborators are grouped in the same group of the colleagues with whom they interact the most. It is likely that such a partition would erase the advantage of Rotating over On-Off, as compared to our simulations where individuals are randomly partitioned.

## Conclusion

In this paper, we simulate SARS-CoV-2 transmission and assess the epidemiological impact of various telecommuting strategies. Our study goes beyond previous work by modeling the fine-grained spreading effects of Sars-Cov2, using real-world contact networks at a workplace, a primary school and a high school. To summarize, our results highlight that (1) when $R_0^{local}$ is moderately high, all the hybrid telecommuting strategies considered reduce it to less than 1, and the choice between them should primarily be done on the basis of practical considerations. (2) To help prevent dissemination of the disease, it is preferable to alternate over longer periods (weekly rather than daily), but the difference is so slight that practical, psychological, and other considerations should determine the alternation time. In future work, it might be

interesting to incorporate the real-life networks as blocks within larger synthetic networks for simulations at larger scales of society.

## Supporting information

**S1 Text. Supporting information.**
(PDF)

**S1 Fig. Primary school graphs extracted from** http://www.sociopatterns.org/wp-content/uploads/2015/09/primaryschool.csv.gz**.** The two days of the trace correspond to Thursday and Friday. Each day is represented by a graph where a node corresponds to an individual, and an edge corresponds one or several face contacts. Edge width corresponds to the number of contacts. Node sizes correspond to weighted degrees. Node colors correspond to known groups which are classes. There size vary between 22 and 27. We observe many contacts between classes of the same grade (e.g. 5A and 5B).
(TIF)

**S2 Fig. Workplace graphs extracted from** http://www.sociopatterns.org/wp-content/uploads/2018/12/tij_InVS15.dat_.gz**.** The trace lasts over two weeks and contains contacts only during working days. Each day is represented by a graph where a node corresponds to an individual, and an edge corresponds one or several face contacts. Edge width corresponds to the number of contacts. Node sizes correspond to weighted degrees. Node colors correspond to known groups which are departments. Their size vary from 2 to 57, most of them contain at most 32 persons.
(TIF)

**S3 Fig. Highschool graphs extracted from** http://www.sociopatterns.org/wp-content/uploads/2015/07/High-School_data_2013.csv.gz**.** The trace lasts during the 5 working days of a week. Each day is represented by a graph where a node corresponds to an individual, and an edge corresponds one or several face contacts. Edge width corresponds to the number of contacts. Node sizes correspond to weighted degrees. Node colors correspond to known groups which are classes. Their size vary from 29 to 44.
(TIF)

**S4 Fig. Random uniform graph.** The trace lasts one day (average graph is identical to Day 1). It is represented by a graph where a node corresponds to an individual, and an edge corresponds one or several face contacts. The graph is calibrated so that its main parameters (total number of nodes, of edges, and of contacts) match those of the high-school average graph: more precisely, each edge is generated by selecting uniformly at random two nodes with one associated contact (rejecting loops and already generated pairs) and each of the remaining contacts is associated to an edge selected uniformly at random among the previously generated edges. Edge width corresponds to the number of contacts that were associated to it. Node sizes correspond to weighted degrees. Node colors correspond to groups which were selected uniformly at random for each node within 9 fixed groups. Their size vary from 30 to 48.
(TIF)

**S5 Fig. Sensitivity of the results to the choice of contact graph.** Panel (c) is identical to Fig 3. We see that regardless of the contact graph, the ranking of the strategies by effectiveness is the same, thus the qualitative results are robust. Note that the quantitative results are also similar from graph to graph, with the exception of the total number of infected people: for the high school contact graph (c), it equals 34.8, whereas for the synthetic random graph (d), it equals 72.3. That happens in spite of the fact that the random graph is calibrated to be the same as the

high school graph in terms of number of nodes, edges, and contacts: thus, the difference is due to the expansion of the random graph, which contrasts with the high school group structure. (TIF)

**S6 Fig. Sensitivity of the results to the choice of contact graph.** Panel (c) is identical to Fig 4. Qualitatively, we see that the order between the curves is the same for all contact graphs and all values of $R_0^{local}$, so that result is robust. The weekly and daily alternations are indistinguishable for this measure. Quantitatively, if we focus on the largest $R_0^{local}$ such that On-Off leads to $R_e <$ 1, we see that it depends significantly on the underlying contact graph: $R_0^{local} = 1.52$ for primary schools, 1.30 for the workplace, 1.38 for the high school, and 1.55 for the random graph. (TIF)

**S7 Fig. Sensitivity of the results to the choice of contact graph.** For all contact networks, we performed a sensitivity analysis of the results of S5 Fig w.r.t. $R_0^{local}$ (or equivalently, to the parameter $p$.) We see that the probability of an outbreak is sensitive to the value of $R_0^{local}$: for example, for On-Off, as $R_0^{local}$ varies from 1 to 1.5, it goes from 21% to 33%. However, it is not so sensitive to the choice of contact graph: when there is no strategy, for the base case $R_0^{local} =$ 1.25 it is around .25 for all graphs. The number of days until an outbreak, around two weeks, is fairly robust and shows little sensitivity to either $R_0^{local}$ or the choice of contact graph. The final number of people infected conditioned on an outbreak is the most sensitive quantity, both to the value of $R_0^{local}$ and to the choice of contact graph. (TIF)

**S8 Fig. Sensitivity of the results to the choice of contact graph.** Numerical data of S5–S7 Figs when $R_0^{local} = 1.25$. (TIF)

**S9 Fig. Sensitivity analysis of the results of Fig 3, for the high school contact network, w.r.t. graph of persistent contacts.** In part (a), we do a sensitivity analysis when we vary the intensity of persistent contacts (baseline: 25%). The baseline case in (a) corresponds to the vertical dotted line, whose intersection with the curves of the strategies gives the values of Fig 3. We see that, the more persistent contacts there are, the worse it is for the epidemic, but that the variation is smooth. Parts (b), (c) and (d) we do a sensitivity analysis in which we vary the *structure* of the persistent contacts graph, while keeping the total number of contacts unchanged. Part (c) is the baseline case and is an identical copy of Fig 3, for ease of comparison. Part (d) takes a complete homogeneous graph for the persistent contacts graph. Part (b) is a construction of what we call a *best friends* graph, constructed in the following two steps: First, each person lists their neighbor by order of decreasing number of contacts, stopping as soon as they reach 25% or their total number of contacts. This creates a directed graph in which many arcs carry 0 contacts. Second, we make it symmetric by putting on each edge {$u$, $v$} the average of the number of contacts on arc($u$, $v$) and on arc ($v$, $u$). We observe that the results are sensitive to the structure. The best friends graph propagates the epidemic the least, the complete graph propagates it the most, and the baseline graph is intermediate. For example, regarding the probability of outbreak, when there is no strategy the probability is 33% for the complete graph, 27% for the baseline graph, and 23% for the best friends graph. Regarding the total number of persons infected when there is an outbreak, when there is no strategy we have 151.8 for the complete graph (the bar actually goes beyond the figure), 34.8 for the baseline graph, and 23.1 for the best friends graph. When the daily On-Off strategy is used, the numbers are 59.7 for the complete graph, 17.4 for the baseline graph, and just 11.8 for the best friends graph. We see that the structure of contacts in the high school, that mostly happen

within well-separated groups, results in a much smaller number of infected people. A further behavioral change in which people reduce the number of people they interact with to just a few best friends, even without reducing their total number of contacts, results in a significant reduction in the number of people infected.
(TIF)

**S10 Fig. Numerical data of S9 Fig for different types of persistent contacts, in the high school contact graph, with a transmission parameter set to have $R_0^{local} = 1.25$ in baseline.** Thus, for the high school contact network, compared to having no strategy, Daily On-Off reduces the reproduction number by $1 - 0.48/0.91 = 47\%$ and Daily Rotation reduces it by $1 - 0.23/0.91 = 75\%$. The improvement of weekly strategies over their daily analog is less than 2%.
(TIF)

**S11 Fig. Sensitivity of S6 Fig if we remove the graph of persistent contacts, which corresponds to a strict curfew.**
(TIF)

**S12 Fig. Sensitivity of S7 Fig if we remove the graph of persistent contacts, which corresponds to a strict curfew.**
(TIF)

**S13 Fig. Sensitivity analysis of the results of Fig 3, for the high school contact network.** In (a), we look at the parameters as a function of $R_0^{local}$, which varies by changing the value of the probability $p$ of symptomatic transmission (baseline $R_0^{local} = 1.25$ corresponding to $p = 0.0035$, and $R_0^{local} = 1$ corresponds to $p = 0.025$). We do not observe a phase transition in which the number of infected people would explode when $R_0^{local}$ becomes greater than 1, but instead, we observe a smooth increase. This is probably due to the small size of the graph (327 nodes), too small to see the theoretical asymptotic behavior as the number of nodes goes to infinity. In (b), we look at the parameters by changing the shape of the super-spreading distribution (gamma of mean 1, baseline shape value 0.1). The baseline case corresponds to the vertical dotted line at $R_0^{local} = 1.25$, whose intersection with the curves of the strategies gives the values of Fig 3.
(TIF)

**S14 Fig. Sensitivity analysis of the results of Fig 3, for the high school contact network, w.r. t.** (a) the difference of infectiousness of an asymptomatic person compared to that of a symptomatic person (baseline: 1/2); here there is a tradeoff in the duration until outbreak, conditioning on existence of an outbreak: when asymptomatic persons are almost not infectious, the epidemic evolution is driven by symptomatic persons, who are only able to contaminate others in the first few days before they isolate, so when outbreaks do happen, they happen more quickly; at the other end of the scale, when most asymptomatic people are just as infectious as symptomatic people, they are infectious for many days but because they are more contagious, they infect people earlier on. (b) the number of days during which a symptomatic individuals continues going to school or work after developing symptoms (baseline: 1 day). The baseline case corresponds to the vertical dotted line, whose intersection with the curves of the strategies gives the values of Fig 3. Part (b) suggests that changing behavior so that a person self-isolates as soon as she develops symptoms is very effective to reduce the dissemination of the epidemic in her contact network.
(TIF)

**S15 Fig. Sensitivity analysis of the results of S5 Fig, for the four contact networks, w.r.t. the probability that an infected person is asymptomatic (baseline: 40%); as expected given that symptomatic individuals isolate, the higher the fraction of asymptomatic persons, the worse the outbreak is.**
(TIF)

**S16 Fig. Sensitivity analysis of the results of Fig 3, for the high school contact network, w.r.t. the SEIR model parameters.** (a) the mean length of the exposed period (baseline: 3.7 days); (b) the shape of the distribution of the exposed period (baseline: 5). The baseline case corresponds to the vertical dotted line, whose intersection with the curves of the strategies gives the values of Fig 3. Unsurprisingly, the longer the exposed period, the more time it takes before 5 people are infected; otherwise the distribution of the exposed period has little impact on the results.
(TIF)

**S17 Fig. Sensitivity analysis of the results of Fig 3, for the high school contact network, w.r.t. the SEIR model parameters.** (a) the mean length of the infectious period (baseline: 9.5 days); (b) the shape of the distribution of the remaining of the infectious period after the first 1.5 days (baseline: 10). The baseline case corresponds to the vertical dotted line, whose intersection with the curves of the strategies gives the values of Fig 3. The variations are monotone and smooth.
(TIF)

**S18 Fig. Sensitivity of S5 Fig to an incomplete input contact network.** The primary school contact network is missing 4% of the children, who did not participate in the Sociopatterns study. Our baseline results for primary schools are therefore on the 4% vertical line. To analyze the effect of removing participants, we remove additional participants uniformly at random, starting from the data with 4% missing and going all the way to 50%. We proceed similarly for high schools (baseline: 14%) and for the workplace (baseline: 30%). We see that, unsurprisingly, removing people reduces the probability of outbreak, reduces the expected final number of people infected conditioned on an outbreak, and increases the number of days until there is an outbreak. Thus, our quantitative results seem to be an underestimate of the situation in the actual contact network. Note that the value of $p$ was not recalibrated for each new network, hence sparser networks have a lower $R$ value.
(TIF)

**S19 Fig. Four simulation runs of epidemic propagation inside the primary school network (similarly to Fig 5).** Among the runs producing an outbreak under no strategy, we selected the first four that produce a median number of infections when we do not implement a strategy, that is 37.
(TIF)

**S20 Fig. Four simulation runs of epidemic propagation inside the workplace network (similarly to Fig 5).** Among the runs producing an outbreak under no strategy, we selected the first four that produce a median number of infections, that is 43.
(TIF)

**S21 Fig. Four simulation runs of epidemic propagation inside the highschool network (similarly to Fig 5).** Among the runs producing an outbreak under no strategy, we selected the first four that produce a median number of infections, that is 26.
(TIF)

**S22 Fig. Four simulation runs of epidemic propagation inside the random uniform graph (similarly to Fig 5).** Among the runs producing an outbreak under no strategy, we selected the first four that produce a median number of infections, that is 47.
(TIF)

## Acknowledgments

We wish to thank the coordinators of the MODCOV19 project for providing interesting references and contacts, and Simon Cauchemez for an inspiring discussion about modeling superspreaders.

## Author Contributions

**Conceptualization:** Simon Mauras, Vincent Cohen-Addad, Guillaume Duboc, Max Dupré la Tour, Paolo Frasca, Claire Mathieu, Lulla Opatowski, Laurent Viennot.

**Funding acquisition:** Claire Mathieu, Lulla Opatowski.

**Investigation:** Simon Mauras, Guillaume Duboc, Max Dupré la Tour.

**Methodology:** Simon Mauras, Vincent Cohen-Addad, Guillaume Duboc, Max Dupré la Tour, Paolo Frasca, Claire Mathieu, Lulla Opatowski, Laurent Viennot.

**Software:** Simon Mauras, Guillaume Duboc, Max Dupré la Tour, Laurent Viennot.

**Supervision:** Claire Mathieu, Lulla Opatowski.

**Visualization:** Laurent Viennot.

**Writing – original draft:** Simon Mauras, Vincent Cohen-Addad, Guillaume Duboc, Max Dupré la Tour, Paolo Frasca, Claire Mathieu, Lulla Opatowski, Laurent Viennot.

**Writing – review & editing:** Simon Mauras, Vincent Cohen-Addad, Guillaume Duboc, Max Dupré la Tour, Paolo Frasca, Claire Mathieu, Lulla Opatowski, Laurent Viennot.

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
