## [Decision Letter · Decision Letter 0]

1 Apr 2021

Dear Mr Mauras,

Thank you very much for submitting your manuscript "Mitigating COVID-19 outbreaks in workplaces and schools by hybrid telecommuting" for consideration at PLOS Computational Biology.

As with all papers reviewed by the journal, your manuscript was reviewed by members of the editorial board and by several independent reviewers. In light of the reviews (below this email), we would like to invite the resubmission of a significantly-revised version that takes into account the reviewers' comments.

We cannot make any decision about publication until we have seen the revised manuscript and your response to the reviewers' comments. Your revised manuscript is also likely to be sent to reviewers for further evaluation.

Sincerely,

Roger Dimitri Kouyos

Associate Editor

PLOS Computational Biology

Tom Britton

Deputy Editor

PLOS Computational Biology

Reviewer's Responses to Questions

**Comments to the Authors:**

Reviewer #1: The paper simulates the transmission of SARS-CoV-2 in three environments: (i) a primary school, (ii) an office, and (iii) a high school. Results show that Rotating strategies dominate the On-Off strategies which in turns dominates the absence of any policy and that in the long run weekly alternation is a bit better than daily alternation, both for On-Off and for Rotating strategies.

The main limitation of the study that does not seem to be acknowledged in the discussion is the lack of age structure in the model. It is established that the epidemic affects differently people of different ages, for example when it comes to the probability of developing asymptomatic infections, (Davies NG, Klepac P, Liu Y, Prem K, Jit M, Eggo RM. Age-dependent effects in the transmission and control of COVID-19 epidemics. Nature medicine. 2020 Aug;26(8):1205-11.) As different age groups are analysed in these networks, these differences should be considered.

A few additional comments:

First line of the abstract, edit the word ‘epidemics’ with ‘epidemic’.

Lines 32-36 introduce a discussion point about the methodology of the work in a larger context and should be moved in the Discussion paragraph.

The caption of Fig.2 reports several epidemiological information about the duration of exposed and infectious periods and symptoms, these are inputs to the model and should all be referenced in the caption as well.

Line 44, change “dissemination of the disease” with “spread of the infection”.

In Methods, subsection contact networks, it should be explained how the authors took into account the missing coverage in the three datasets (4% children, 30% employees, 14% students) as this means that in reality individuals got in contact with a larger number of people, affecting the R0 in the study analysis. If this was not taken into account, the authors should mention it in the study limitations.

Lines 63-72 describe the method and should be incorporated in the Strategy section. Each strategy would benefit to be named and presented in a table and then referred when presenting result.

Reviewer #2: The paper provides an intersting study for the outbreak of Covid-19 epidemic in three real life small contact networks (two schools and one workplace in France), considering two NPI strategies (rotating, with two different time-lags and on-off).

It is well written, theoretically grounded, and easy to follow. As such I only have a few remarks that the authors may want to adress in the revision.

1) The analysis has "local" value. How could be in principle the results be extrapoleted population to give more general guidelines to policy makers on whether or not follow the "best" rotating strategy. In other words, looking at the national available infection curves (more than one year now) and the measures implemented in France (for instance in schools), how adherent are the results of the paper on what happend so far? The "retrospective" value of the paper would considerably enhance the analysis.

2) In the currect situation there are two main facts impacting on the evolution of the pandemic: (i) the presence of variants (especially the "English" one) , that increase trasmissibility (and hence the basic R0) from R0=2.5 to about R0=3.3 (with no restrictions). The computations done in the paper seem not to considere this, if I well understood, (ii) the presence of vaccination - even if in France is still not so massive: is it possible to incorporate in the contact network a vaccination parameters that decreases the susceptibles linearly in time?

3) It is clear (and can be proven mathematically) that an on-off stategy is always worst than an off-on strategy, at parity of social-economic costs (that rougly depen only on the closure time interval). The initial amount of infected people is really important and and openings should be suggested only when this number is sufficiently low. I don't see this reasoning in the paper.

4) Could you please give some more details on the stochastic simulations (Gillespy?) and on the underlying stochastic process chain? For individuals (or groups) interactions normally a Markov chain is defined. The overall master model is in general not reducible (marginalizable) at the first moment to get standard epidemiologic detreministic models (like SEIR). Is this the reason of the discrepancy quoted in thye method section?

5) In Fig. 3 (medium) the difference in days between no-strategy and full telecummuting is quite low. Are there any reason?

6) In Fig. 4, the variable p appears, not defined previously.

**Have all data underlying the figures and results presented in the manuscript been provided?**

Reviewer #1: None

Reviewer #2: Yes

PLOS authors have the option to publish the peer review history of their article (what does this mean?). If published, this will include your full peer review and any attached files.

Reviewer #1: No

Reviewer #2: No
---

## [Decision Letter · Decision Letter 1]

10 Jul 2021

Dear Mr Mauras,

We are pleased to inform you that your manuscript 'Mitigating COVID-19 outbreaks in workplaces and schools by hybrid telecommuting' has been provisionally accepted for publication in PLOS Computational Biology.

Best regards,

Roger Dimitri Kouyos

Associate Editor

PLOS Computational Biology

Tom Britton

Deputy Editor

PLOS Computational Biology

Reviewer's Responses to Questions

**Comments to the Authors:**

Reviewer #1: My previous comments and the ones from the other reviewer have been addressed to my satisfaction. I believe the paper to be worth of publication.

Reviewer #2: The authors have taken seriously all concerns raised in the previous round of review including

1) the retrospective analysis (no data),

2) variants and vaccinations (two paragraphes added. At this end the authors could consider to include a reference on the compromize closures/vaccination strength, for instance the recent paper "Modeling vaccination rollouts, SARS-CoV-2 variants and the requirement for non-pharmaceutical interventions in Italy", Nature >Medicine, 2021.

3) On-off/off-on (symmetry) stochastic simulation (paragraph added).

Conclusion:

Despite the recognized limitations (as in all papers) I think that the paper is strong, well written and thoroughly addresses the important topic of rotation/on off strategies in the given set-up (schools/workplaces). As such I strongly suggest publication.

**Have the authors made all data and (if applicable) computational code underlying the findings in their manuscript fully available?**

Reviewer #1: None

Reviewer #2: None

PLOS authors have the option to publish the peer review history of their article (what does this mean?). If published, this will include your full peer review and any attached files.

Reviewer #1: No

Reviewer #2: No

---

## [Editor Report · Acceptance letter]

5 Aug 2021

PCOMPBIOL-D-21-00219R1 

Mitigating COVID-19 outbreaks in workplaces and schools by hybrid telecommuting

Dear Dr Mauras,

I am pleased to inform you that your manuscript has been formally accepted for publication in PLOS Computational Biology. Your manuscript is now with our production department and you will be notified of the publication date in due course.

With kind regards,

Agnes Pap
